# SOPSeg: Prompt-based Small Object Instance Segmentation in Remote Sensing Imagery

## Abstract

Extracting small objects from remote sensing imagery plays a vital role in various applications, including urban planning, environmental monitoring, and disaster management. While current research primarily focuses on small object detection, instance segmentation for small objects remains underexplored, with no dedicated datasets available. This gap stems from the technical challenges and high costs of pixel-level annotation for small objects. While the Segment Anything Model (SAM) demonstrates impressive zero-shot generalization, its performance on small-object segmentation deteriorates significantly, largely due to the coarse 1/16 feature resolution that causes severe loss of fine spatial details. To this end, we propose SOPSeg, a prompt-based framework specifically designed for small object segmentation in remote sensing imagery. It incorporates a region-adaptive magnification strategy to preserve fine-grained details, and employs a customized decoder that integrates edge prediction and progressive refinement for accurate boundary delineation. Moreover, we introduce a novel prompting mechanism tailored to the oriented bounding boxes widely adopted in remote sensing applications. SOPSeg outperforms existing methods in small object segmentation and facilitates efficient dataset construction for remote sensing tasks. We further construct a comprehensive small object instance segmentation dataset based on SODA-A, and will release both the model and dataset to support future research.

## 1 Introduction

Remote sensing imagery plays a critical role in a wide range of real-world applications, including urban planning, environmental monitoring, and precision agriculture. Among the targets of interest in these applications, small objects such as vehicles, plane, and ships typically occupy no more than 32×32 pixels in high-resolution imagery, yet they convey essential semantic and operational information for downstream tasks. Consequently, accurately extracting small objects is of great importance, but remains a highly challenging task due to their limited size and complex visual characteristics.

Benchmarks such as SODA-A Cheng et al. (2023) have significantly advanced small object detection in remote sensing imagery. However, they provide only bounding box annotations, which constrain models to coarse localization and fail to capture precise object shapes. Consequently, most existing works focus on object detection rather than instance segmentation, limiting fine-grained scene understanding.

Instance segmentation for small objects remains largely underexplored, primarily due to the lack of suitable datasets. Constructing such datasets is highly labor-intensive, error-prone, and requires substantial domain expertise. Although the Segment Anything Model (SAM) Kirillov et al. (2023), trained on over one billion masks, demonstrates strong zero-shot generalization capabilities, its direct application to high-resolution remote sensing imagery leads to notable performance degradation for small objects. We attribute this limitation to the architectural design of SAM: its vision transformer encoder downsamples input images to 1/16 of the original resolution to reduce computational cost. While effective for typical object sizes, this aggressive downsampling results in the loss of fine-grained details that are critical for accurately identifying small targets.

To this end, we propose **SOPSeg** (**S**mall **O**bject **P**rompted **Seg**mentation), a novel framework that adapts SAM for robust small-object instance segmentation in remote sensing imagery. Our approach

introduces three key innovations: (1) Region-adaptive magnification, which adaptively crops and resizes object regions to preserve fine details lost in downsampling, enabling accurate segmentation of small instances with minimal overhead; (2) An edge-aware decoder, which integrates boundary prediction and progressive multi-scale refinement to produce sharper and more accurate object masks; (3) An oriented prompting mechanism, which enables the use of rotated bounding boxes common in aerial imagery, improving SAM's ability to handle objects at arbitrary orientations.

We train and validate SOPSeg on the iSAID dataset Waqas Zamir et al. (2019), selecting 7 out of 15 categories that best represent small object challenges in remote sensing imagery. Generalization ability is further evaluated on the NWPU-VHR10 Su et al. (2019) and SAT-MTB Li et al. (2023) benchmarks. Experimental results show that SOPSeg significantly outperforms the original SAM and other prompt-based segmentation methods across all datasets.

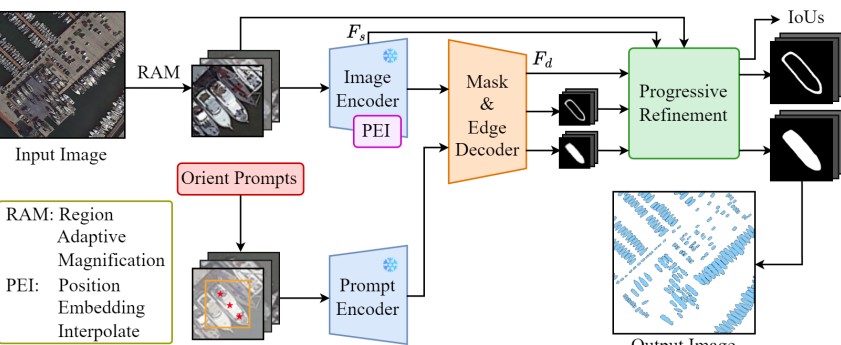

Figure 1: Overview of the SOPSeg Framework. The input remote sensing image is first proceseed by RAM to get multiple patches of uniform size. These patches are then fed into the SAM image encoder, where Position Embedding Interpolation (PEI) is applied to support arbitrary input sizes. An oriented prompt, consisting of a horizontal bounding box and three keypoints aligned with the orientation of the object, is encoded via the SAM prompt encoder to guide segmentation. Coarse masks and edges are generated and refined progressively to yield accurate high-resolution segmentation. $\mathbf{F_d}$ and $\mathbf{F_s}$ denote deep image features from decoder and shallow image features from image encoder.

To demonstrate its practical utility, we further apply SOPSeg to assist in constructing a small object instance segmentation dataset. Specifically, we automatically generate approximately 709k instance masks for small objects based on images and oriented bounding boxes from the SODA-A dataset, followed by manual filtering to remove a small number of abnormal annotations. The resulting dataset **ReSOS** (**Re**mote Sensing **S**mall **O**bject **S**egmentation), represents the first large-scale instance segmentation benchmark focused on small objects in remote sensing imagery. We plan to publicly release both the model and the dataset to provide training and evaluation resources for future research on small object analysis.

In summary, our contributions are threefold:

- We propose SOPSeg, a prompt-based framework that adapts SAM for small object instance segmentation, integrating region-adaptive magnification and edge-aware refinement decoding to enhance mask accuracy.
- We develop an oriented prompting mechanism enabling accurate segmentation of objects at arbitrary orientations.
- We construct and release ReSOS dataset, the first large-scale instance segmentation dataset specifically designed for small objects in remote sensing. It contains pixel-level annotations for over 709k instances and aims to support future research.

## 2 RELATED WORK

**Small Object Detection and Segmentation in Remote Sensing.** Small object analysis in remote sensing has attracted significant research interest due to its practical importance. Early de-

tection methods relied on hand-crafted features and traditional machine learning classifiers Cheng & Han (2016). However, these approaches struggled with the complex backgrounds and varying scales characteristic of aerial imagery. Ding Ding et al. (2019) proposed a rotation-invariant detector specifically designed for aerial images, while Yang Yang et al. (2019) introduced SCRDet to handle the multi-scale and multi-orientation challenges. Recent methods have focused on feature enhancement strategies. RMSIN Liu et al. (2024) employs interaction modules to effectively capture complex spatial scales and orientations for accurate segmentation in remote sensing imagery. For instance, FCOS-RS Li et al. (2020) adapts the anchor-free FCOS detector for remote sensing by incorporating multi-scale feature fusion. Similarly, Oriented R-CNN Xie et al. (2021) extends Faster R-CNN Ren et al. (2016) with oriented region proposals to better capture arbitrarily oriented objects. Despite progress in detection, instance segmentation of small objects remains largely unexplored. The few existing works primarily focus on specific object categories. Zhang Zhang et al. (2017) developed a ship instance segmentation method using polar coordinates, while Zhao Zhao et al. (2021) proposed building extraction techniques. UGBS Yang et al. (2024) explored interactive user guidance mechanisms to achieve more accurate building segmentation from high-resolution remote sensing images, demonstrating the potential of human-in-the-loop approaches. However, these category-specific approaches do not generalize to diverse small objects. The scarcity of segmentation methods stems from the lack of appropriate datasets and the inherent difficulty of obtaining pixel-level annotations for tiny objects.

**Segment Anything Model and Applications.** The Segment Anything Model (SAM) Kirillov et al. (2023) represents a paradigm shift in image segmentation through its foundation model approach. SAM's versatility stems from its flexible prompting mechanism. Users can specify objects of interest through points, bounding boxes, or coarse masks, enabling interactive segmentation workflows. Recent works have explored SAM's potential in remote sensing applications. SAMRS Wang et al. (2023) leverages SAM to automatically convert object detection datasets into instance segmentation datasets, demonstrating its utility for large-scale annotation tasks. SAM2 Ravi et al. (2024) enhanced segmentation accuracy on both images and videos. RSPrompter Chen et al. (2023a) introduces auxiliary prompts specifically designed for remote sensing imagery to improve SAM's performance. SAM-Adapter Chen et al. (2023b) proposes lightweight adapters to adapt SAM for domain-specific tasks while preserving its zero-shot capabilities. ROS-SAM Shan et al. (2025) specifically targets moving object segmentation in remote sensing videos by leveraging LoRA-based adaptation and a context-aware decoder. It primarily focuses on objects with sufficient motion patterns rather than addressing the challenges of small object segmentation. HQ-SAM Ke et al. (2023) addresses the issue of coarse mask boundaries in the original SAM by introducing a learnable High-Quality Output Token. Matting Anything Li et al. (2024) extends SAM to the image matting task by predicting precise alpha channels for objects with complex boundaries. Nevertheless, constrained by the resolution of low-level features, existing approaches exhibit limited performance in small object segmentation — a key challenge that this study seeks to systematically tackle.

## 3 METHODOLOGY

To bridge the gap between generic segmentation models and the unique demands of small object segmentation in remote sensing, we propose SOPSeg, a prompt-based framework that introduces three key improvements over SAM: a region-adaptive magnification strategy, an oriented prompt mechanism, and an enhanced decoder with integrated edge prediction. The overall architecture is illustrated in Figure 1.

### 3.1 REGION-ADAPTIVE MAGNIFICATION STRATEGY

The core challenge in small object segmentation lies in preserving spatial details during feature extraction. SAM's vanilla image encoder processes images at a fixed resolution, downsampling features to 1/16 of the original size. For small objects occupying only $32 \times 32$ pixels, this results in feature representations of merely $2 \times 2$ pixels, causing severe information loss.

Our region-adaptive magnification strategy addresses this limitation through adaptive region extraction and resizing. Given an input image and the bounding box $b = (x, y, w, h)$ of a instance, we extract a square local region with boundaries $(x_s, y_s, S, S)$, where the size $S$ is determined by the object size $d = \max(w, h)$. Each region is then resized to a fixed resolution of $S_{\text{in}} \times S_{\text{in}}$ before

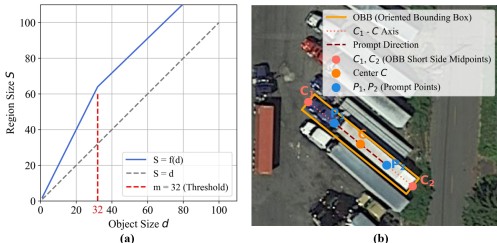

Figure 2: (a) The relationship between the extracted region size $S$ and object dimension $d = max(w, h)$. (b) Illustration of the oriented prompting mechanism for rotated objects. The final prompt points include $P_1$, $C$, and $P_2$.

being fed into the model. The relationship between $S$ and $d$ is formulated as:

$$S = \begin{cases} k_0 \cdot d, & \text{if } d < m \\ k \cdot d + (k_0 - k) \cdot m, & \text{if } d \geq m \end{cases} \quad (1)$$

where: $k = \frac{S_{\max} - k_0 \cdot m}{S_{\max} - m}$ and satisfies the boundary condition $S = S_{\max}$ when $d = S_{\max}$. A visual illustration is provided in Fig. 2(a). We empirically set $m = 32$, $k_0 = 2$, and $S_{\max} = 1024$. Here, $k_0$ represents the initial region expand factor for objects smaller than the threshold $m$. Since all regions are resized to a fixed input size $S_{\text{in}}$, the object magnification is $S_{\text{in}}/S$. Smaller $S$ yields larger magnification, which benefits small objects by enhancing fine details. For objects larger than the threshold, appropriately reducing surrounding context increases their magnification while still retaining essential context. This adaptive design balances detail preservation for small instances and contextual integrity for large ones.

We compute the top-left coordinates $(x_s, y_s)$ based on the desired region size:

$$\begin{bmatrix} x_s \\ y_s \end{bmatrix} = \begin{bmatrix} x \\ y \end{bmatrix} - \begin{bmatrix} a_x(S - w) \\ a_y(S - h) \end{bmatrix} \quad (2)$$

where $a_x$ and $a_y$ control the object's position within the extracted region. During training, we set $a_x, a_y \in [0.3, 0.7]$ randomly to improve generalization capability, ensuring objects appear at various positions rather than always centered.

SAM's default $1024 \times 1024$ input resolution is designed for processing entire images containing objects of various sizes. However, when focusing on small objects through region extraction, this high resolution becomes computationally wasteful—most pixels represent irrelevant background rather than the target object. Thus we set the input size to $S_{\text{in}} = 256$, which preserves sufficient detail for accurate segmentation while significantly reducing computational overhead. For instance, if a small object originally spans $32 \times 32$ pixels, the extracted region size $S = 64$, and the object is effectively magnified by a factor of $S_{\text{in}}/S = 4$.

**Position Embedding Interpolation.** Since SAM's positional embedding weights are input-size dependent, the pretrained embeddings trained on $1024 \times 1024$ inputs cannot be directly reused. We address this through bilinear interpolation:

$$\text{PE}_{\text{target}} = \text{Interpolate}(\text{PE}_{1024}, S_{\text{in}}, S_{\text{in}}) \quad (3)$$

where $S_{\text{in}} = 256$ for small object processing. This interpolation preserves the relative spatial patterns while adapting to the new resolution.

The combination of region extraction, magnification, and reduced input resolution creates an efficient pipeline: small objects are first magnified to an adequate size, then processed at lower input size without losing critical details.

### 3.2 ORIENTED PROMPT MECHANISM

Remote sensing objects frequently appear at arbitrary orientations, posing challenges for standard segmentation models. To this end, we propose a strategy that encodes object orientation using

strategically placed points, thereby enabling the original SAM—designed for axis-aligned bounding boxes—to effectively handle rotated objects in aerial imagery, all without requiring any modifications to its architecture.

For each oriented bounding box, we extract three key geometric points: the geometric center $C$, and the midpoints $C_1$ and $C_2$ of the two shorter sides. The line segment $\overline{C_1 C_2}$ naturally defines the object's principal axis. Apart from $C$, we generate two prompt points along the principal axis:

$$P_1 = \frac{C + C_1}{2}, \quad P_2 = \frac{C + C_2}{2} \tag{4}$$

Fig. 2(b) illustrates our oriented prompt mechanism on a real example from aerial imagery.

These points encode both spatial and directional information:

- The vector $\overrightarrow{P_1 P_2}$ implicitly represents the object's orientation.
- The distance $||P_1 - P_2||$ correlates with the object's length along its principal axis.
- All points remain well within object boundaries, ensuring reliable prompting.

The three points $(P_1, C, P_2)$ are directly processed through SAM's pretrained point encoder:

$$E_{\text{points}} = \text{PointEncoder}([P_1, C, P_2]) \tag{5}$$

Combined with the horizontal bounding box prompt, this provides comprehensive spatial guidance:

$$E_{\text{prompt}} = \text{Concat}(\text{BoxEncoder}(b_{\text{horizontal}}), E_{\text{points}}) \tag{6}$$

This design maintains full compatibility with SAM's pretrained weights while effectively handling arbitrary orientations. The approach is particularly well-suited for elongated objects prevalent in remote sensing, such as vehicles and ships, where the short-edge midpoints naturally capture the object's dominant direction.

### 3.3 ENHANCED DECODER WITH EDGE PREDICTION

Despite the region magnification strategy, small objects in remote sensing imagery still suffer from boundary ambiguity due to complex backgrounds. We introduce an auxiliary edge prediction path and progressive refinement, enhancing fine-grained delineation of small instances.

**Stage 1: Parallel Edge Prediction.** We augment the SAM decoder with a learnable edge token $\mathbf{T}_{\text{edge}}$, which collaborates with the original mask tokens $\mathbf{T}_{\text{mask}}$ to capture boundary-specific information. The $\mathbf{T}_{\text{edge}}$, $\mathbf{T}_{\text{mask}}$, and prompt tokens perform bidirectional attention with the image features, resulting in updated representations: $\mathbf{T}_{\text{edge}}^{(1)}$, $\mathbf{T}_{\text{mask}}^{(1)}$, and $\mathbf{F}_d$. These are then used to generate two parallel outputs:

$$\mathbf{M}_0 = \text{MLP}_{\text{mask}}(\mathbf{T}_{\text{mask}}^{(1)}) \cdot O_{\text{mask}}(\mathbf{F}_d) \tag{7}$$

$$\mathbf{E}_0 = \text{MLP}_{\text{edge}}(\mathbf{T}_{\text{edge}}^{(1)}) \cdot O_{\text{edge}}(\mathbf{F}_d) \tag{8}$$

Here, $\text{MLP}_{\text{edge}}$ and $O_{\text{edge}}$ follow the same architectural design as the mask prediction modules in SAM. $\mathbf{M}_0$ and $\mathbf{E}_0$ denote the initial mask and edge predictions at a resolution of 1/4 input size.

**Stage 2: Progressive Refinement.** Initial predictions capture basic structure but lack fine details critical for small objects. We employ multi-scale refinement that gradually improves both masks and edges through iterative processing. The refinement takes four inputs: deep image features $\mathbf{F}_d$ after attention from the decoder, shallow features $\mathbf{F}_s$ from the image encoder, the original image $\mathbf{I}$, and the initial predictions $\mathbf{P}_0 = [\mathbf{M}_0; \mathbf{E}_0]$ from Stage 1.

Both the shallow and deep image features first undergo $2\times$ upsampling and channel dimension reduction mapping for efficient processing. The shallow features $\mathbf{F}_{16s}$ ($\mathbf{F}_s$) are processed through convolution, normalization, and $2\times$ upsampling to produce $\mathbf{F}_{8s}$. Similarly, decoder features $\mathbf{F}_{16d}$ ($\mathbf{F}_d$) are mapped and upscaled to $\mathbf{F}_{8d}$.

**Multi-Scale Refinement.** The refinement operates across three spatial scales: $1/8 \rightarrow 1/4 \rightarrow 1/2 \rightarrow 1/1$, progressively enhancing both mask and edge predictions.

- **Scale 1/8 to 1/4.** We concatenate the upsampled features $\mathbf{F}_{8s}$ and $\mathbf{F}_{8d}$ with the downsampled image $\mathbf{I}_8$ and predictions $\mathbf{P}_8$ downsampled from $\mathbf{P}_0$, and pass them through the first residual refinement block $\mathcal{R}_1$:

$$\mathbf{X}_4 = \mathcal{R}_1([\mathbf{F}_{8s}; \mathbf{F}_{8d}; \mathbf{I}_8; \mathbf{P}_8]) \tag{9}$$

  The refined feature $\mathbf{X}_4$ is then mapped to updated predictions $\mathbf{P}_4 = [\mathbf{M}_4; \mathbf{E}_4]$ via the output head $\phi_1$:

$$\mathbf{P}_4 = \phi_1(\mathbf{X}_4) \tag{10}$$

- **Scales 1/4 to 1/2 to 1/1.** We apply the same refinement pattern iteratively. At each scale $i \in \{4, 2\}$, we use a residual refinement block $\mathcal{R}_j$ and an output head $\phi_j$ to generate updated predictions:

$$\mathbf{X}_{i/2} = \mathcal{R}_j([\mathbf{X}_i; \mathbf{I}_i; \mathbf{P}_i]), \quad \mathbf{P}_{i/2} = \phi_j(\mathbf{X}_{i/2}) \tag{11}$$

  where $\mathbf{I}_i$ is the image downsampled to resolution $1/i$, while $\mathbf{P}_i$ is the output from last iteration.

- **IoU Prediction.** At the final stage, the refined feature $\mathbf{X}_1$ is also used to predict the mask quality score $p_{\text{iou}}$ via a lightweight head consisting of a convolutional layer, ReLU activation, adaptive average pooling, and linear projection:

$$p_{\text{iou}} = \text{IoU}(\mathbf{X}_1) \tag{12}$$

**Optimization Objective.** As a whole, we adopt a multi-task loss function that jointly supervises mask prediction, edge localization, and mask quality estimation:

$$\mathcal{L} = \sum_{i \in \{1,2,4\}} \left( \mathcal{L}_{\text{mask}}^i + \mathcal{L}_{\text{edge}}^i \right) + \lambda_{\text{iou}} \mathcal{L}_{\text{iou}} \tag{13}$$

Here, both $\mathcal{L}_{\text{mask}}^i$ and $\mathcal{L}_{\text{edge}}^i$ are composed of a sum of Binary Cross-Entropy (BCE) and DICE Milletari et al. (2016) losses between the predicted outputs and the ground truth at scale $i$. The ground-truth edge map is derived from the binary mask annotations and smoothed using a $3 \times 3$ Gaussian filter to mitigate aliasing artifacts.

The term $\mathcal{L}_{\text{iou}}$ employs a Smooth L1 loss between the predicted IoU score and the actual IoU computed from the original-resolution mask, guiding the model to produce accurate mask quality estimations. The hyperparameter $\lambda_{\text{iou}}$ balances the contributions of different loss components. In our implementation, we set $\lambda_{\text{iou}} = 5.0$.

## 4 EXPERIMENTS

### 4.1 EXPERIMENTAL SETUP

**Datasets.** We conduct prompted instance segmentation experiments —where the bounding box is provided and the corresponding instance mask is predicted— on three representative remote sensing datasets: iSAID Waqas Zamir et al. (2019), NWPU-VHR10 Su et al. (2019), and SAT-MTB Li et al. (2023). The iSAID dataset is used for both training and evaluation, while NWPU-VHR10 and SAT-MTB serve as benchmarks for assessing the generalization ability of our method.

To focus on typical small object categories prevalent in remote sensing scenarios, we select specific classes from each dataset. For iSAID, we include: ship, plane, helicopter, small vehicle, large vehicle, storage tank, and swimming pool. For NWPU-VHR10, we consider: ship, airplane, vehicle, and storage tank. For SAT-MTB, we evaluate on ship and plane.

**Evaluation Metrics.** We evaluate segmentation performance using mean Intersection over Union (mIoU) and boundary IoU (BIoU) Cheng et al. (2021) across all instances.

**Implementation Details.** We use the image encoder and prompt encoder from SAM-Large and freeze their parameters during training. The enhanced decoder, initialized from SAM-Large weights, trained with learning rates of $5 \times 10^{-5}$. The progressive refinement module is trained from scratch with learning rates of $1 \times 10^{-3}$. The model is trained for 32 epochs using the AdamW optimizer and a cosine annealing learning rate schedule, with a batch size of 24.

Table 1: Comparison of IoU (%) for different methods on the iSAID dataset. We report GFLOPs (of 10 instance on a image), model parameters, and per-class IoU. Abbreviations: UP2(2× image upsampling), ST (Storage Tank), LV (Large Vehicle), SV (Small Vehicle), HC (Helicopter), SP (Swimming Pool).

| Method | GFLOPs | Params | Per-class IoU (%) | | | | | | | mIoU |
|---|---|---|---|---|---|---|---|---|---|---|
| | | | Ship | ST | LV | SV | HC | SP | Plane | |
| SAM | 1342 | 308M | 79.86 | 74.62 | 75.53 | 62.14 | 61.52 | 72.77 | 75.13 | 71.65 |
| SAM Up2 | 5368 | 308M | 81.63 | 79.94 | 78.93 | 73.64 | 60.85 | 76.24 | 78.16 | 75.63 |
| SAM Up4 | 21472 | 308M | 82.93 | 81.69 | 80.36 | 77.59 | 62.58 | 79.58 | 79.74 | 77.78 |
| HQ-SAM | 1376 | 309.3M | 79.20 | 72.73 | 74.50 | 59.06 | 61.22 | 71.65 | 76.94 | 70.76 |
| ROS-SAM | 1594 | 359.7M | 81.61 | 81.41 | 78.94 | 72.35 | 61.31 | 79.10 | 76.52 | 75.89 |
| UGBS | 2172 | **79.4M** | 84.82 | 86.85 | 85.06 | 81.77 | 66.04 | 82.69 | 78.83 | 80.87 |
| **SOPSeg** | **1244** | 311M | **87.14** | **88.54** | **87.23** | **85.28** | **67.55** | **84.34** | **80.63** | **82.96** |

## 4.2 COMPARISON WITH OTHER METHODS

Table 1 compares SOPSeg with several representative prompt-based segmentation methods fine-tuned on the iSAID dataset. We include **SAM** Kirillov et al. (2023) as the foundational interactive segmentation model, and additionally report its performance under 2× and 4× image upsampling. For SAM enhancement approaches, we consider **HQ-SAM** Ke et al. (2023) and **ROS-SAM** Shan et al. (2025). For CNN-based prompt segmentation method in remote sensing, we compare with **UGBS** Yang et al. (2024). Since UGBS relies on one instance region as input, we adopt our proposed RAM strategy to extract the surrounding regions, enabling a fair evaluation. The results show that SOPSeg outperforms all competing methods across all categories, while incurring the lowest computational overhead, requiring only 3M additional parameters beyond the SAM backbone.

The figure 3 shows visualization results of small object prompt-based segmentation on the iSAID dataset, with all instances overlaid. Due to aggressive feature downsampling, SAM suffers from object adhesion, especially on small targets like cars. ROS-SAM and UGBS partially alleviate this issue but still struggle with boundary precision and object separation. In the first row, ROS-SAM also incorrectly segments non-aircraft regions. Our method accurately preserves object shapes across various scenes, with clear boundaries and well-separated instances, achieving results closest to the ground truth.

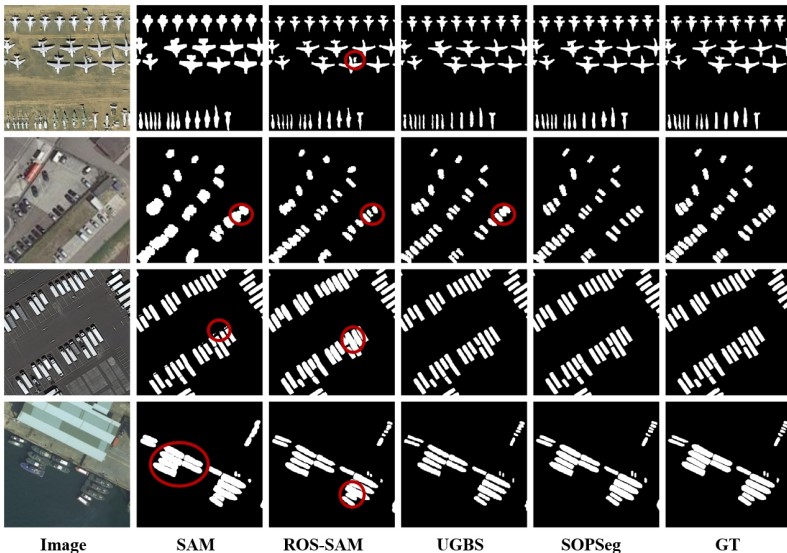

Figure 3: Visualization results of small object segmentation methods on iSAID dataset.

**Generalization testing.** Table 2 evaluates method generalization on NWPU-VHR10 and SAT-MTB datasets. SOPSeg consistently outperforms all baselines. Compared with the strongest baseline

UGBS, SOPSeg yields larger improvements on SAT-MTB than NWPU, likely due to its stronger robustness to cross-dataset distribution shifts.

Table 2: Generalization results on NWPU and SAT-MTB datasets.

| Method | NWPU | | SAT-MTB | |
|--------|------|------|---------|------|
| | IoU | BIoU | IoU | BIoU |
| SAM | 78.70 | 68.56 | 58.72 | 57.49 |
| HQ-SAM | 79.47 | 69.46 | 58.57 | 57.4 |
| ROS-SAM | 82.84 | 75.09 | 68.43 | 67.26 |
| UGBS | 86.13 | 79.33 | 70.32 | 69.54 |
| **SOPSeg** | **86.55** | **80.49** | **73.38** | **72.56** |

Table 3: Evaluation on different decoder architectures.

| Method | IoU | BIoU |
|--------|------|------|
| SAM | 84.17 | 80.54 |
| MatAnything | 84.74 | 81.02 |
| HQ-SAM | 85.06 | 81.62 |
| **Our Decoder** | **85.38** | **81.98** |

Table 4: Ablation study results on the iSAID dataset. RAM: Region-Adaptive Magnification; OPM: Oriented Prompt Mechanism; EDE: Enhanced Decoder with Edge Prediction. Abbreviations: ST (Storage Tank), LV (Large Vehicle), SV (Small Vehicle), SP (Swimming Pool).

| Method | Ship | ST | LV | SV | Helicopter | SP | Plane | mIoU |
|--------|------|------|------|------|-----------|------|-------|------|
| Base | 79.86 | 74.62 | 75.53 | 62.14 | 61.52 | 72.77 | 75.13 | 71.65 |
| +RAM | 83.43 | 85.90 | 83.08 | 79.46 | 64.91 | 80.45 | 78.59 | 79.40 |
| +RAM+OPM | 85.64 | 88.17 | 86.02 | 83.98 | 65.33 | 84.22 | 78.47 | 81.69 |
| **+RAM+OPM+EDE** | **87.14** | **88.54** | **87.23** | **85.28** | **67.55** | **84.34** | **80.63** | **82.96** |

## 4.3 ABLATION STUDY

**Effectiveness of Different Modules.** Table 4 presents the ablation results on the iSAID dataset. We progressively incorporate each component of the SOPSeg framework into a baseline model, which fine-tunes the original SAM decoder using horizontal box prompts. Adding the Region-Adaptive Magnification (RAM) module improves performance by 7.84% mIoU. The RAM module benefits small vehicle and storage tank, where spatial details are often lost during standard downsampling. The oriented prompt mechanism adds 2.29% over the RAM-only configuration. Finally, our enhanced decoder contributes an additional 1.27% improvement. This gain is more evident for classes like plane and helicopter, which exhibit complex boundaries and fine structural details. These results demonstrate that each module brings consistent performance gains, and their combination yields the best overall segmentation performance.

**Decoder Component Analysis.** As shown in Table 3, we evaluate different decoder designs by replacing our decoder with various alternatives. Our enhanced decoder achieves the best overall performance, outperforming the original SAM decoder by 1.44% in BIoU and 1.21% in IoU. These results demonstrate that incorporating edge prediction effectively preserves fine-grained details and improves boundary accuracy for small object segmentation.

**Input Resolution Analysis.** Figure 4 analyzes the impact of different input sizes on both segmentation performance and computational efficiency. We evaluate four input resolutions—128, 256, 384, and 512—to determine the optimal configuration for small object segmentation.

Figure 4(a) shows the per-category performance across different input sizes. An input resolution of 256 achieves the best overall performance across most categories, with particularly strong results for ship, swimming pool, and small vehicle. In contrast, an input resolution of 128 significantly degrades performance for categories like helicopter and plane, which rely heavily on fine-grained spatial details for precise boundary delineation. Interestingly, storage tank performs best at 128 resolution, possibly due to its inherently simple and compact structure, which may become over-smoothed or misrepresented when additional detail is introduced at higher resolutions. Larger input sizes, such as 384 and 512, do not lead to proportional improvements over 256. Although marginal improvements are observed in the plane category, these come at the cost of significantly higher computational demands.

Table 5: IoU results under different $k_0$ settings and magnification strategies in RAM across instance sizes. Abbreviation: Up2($2\times$ image upsampling).

|  | Tiny <16 | Small 16~32 | Middle >32 | All |
|---|---|---|---|---|
| $k_0 = 1.2$ | 79.7 | 85.1 | 86.7 | 84.7 |
| $k_0 = 2$ | **80.7** | **85.7** | **87.3** | **85.4** |
| $k_0 = 4$ | 79.9 | 84.8 | 86.9 | 84.8 |
| w/o adaptive, Up2 | 77.4 | 84.5 | 86.8 | 84.1 |
| w/o adaptive, Up4 | 78.7 | 84.9 | 86.9 | 84.6 |
|  | (+1.3) | (+0.4) | (+0.1) | (+0.5) |
| w/ adaptive | **80.7** | **85.7** | **87.3** | **85.4** |
|  | **(+2.0)** | **(+0.8)** | **(+0.4)** | **(+0.8)** |

Figure 4: The impact of input size on (a) class-wise IoU and (b) mean class IoU and computational cost.

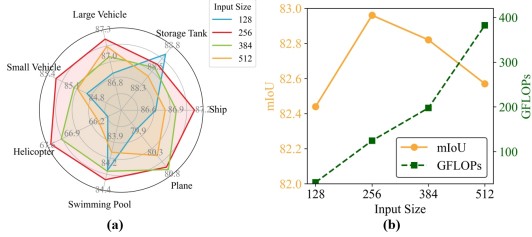

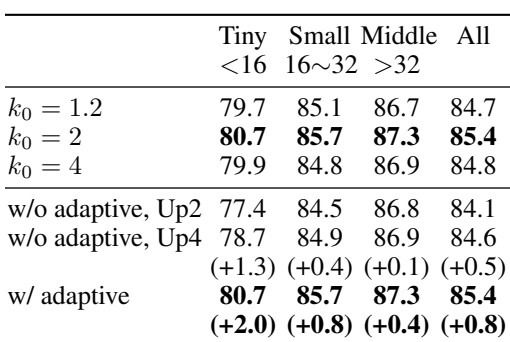

Figure 4(b) presents the computational analysis. The mIoU curve peaks at a resolution of 256, confirming it as the optimal choice in terms of accuracy. Meanwhile, GFLOPs (for one instance) increase dramatically with higher resolution.

**Effect of Magnification Strategies.** As shown in Table 5, we further divide instances into three groups according to their bounding-box size, defined by the maximum side length. Smaller $k_0$ values correspond to higher magnification but reduced context. A moderate setting ($k_0 = 2$) achieves the best balance. The lower part in Table 5 shows that adaptive magnification consistently achieves the highest IoU across all size ranges, with particularly notable improvements for Tiny and Small objects.

### 4.4 EVALUATION ON THE CONSTRUCTED DATASET

Leveraging the strong small object segmentation capability of the proposed SOPSeg, combined with manual filtering, we constructed the **ReSOS** (**Re**mote Sensing **S**mall **O**bject **S**egmentation) dataset based on images and oriented bounding boxes from SODA-A Cheng et al. (2023), containing pixel-level annotations for over 709k instances. This dataset provides a solid foundation for the evaluation and advancement of small object segmentation techniques. Due to page constraints, a detailed description of the dataset is provided in the appendix.

Table 6: Comparison of instance segmentation results (AP, %) on our constructed dataset. Abbreviations: ST (Storage Tank), LV (Large Vehicle), SV (Small Vehicle), SP (Swimming Pool).

| Method | Plane | Helicopter | SV | LV | Ship | Container | ST | SP | AP |
|---|---|---|---|---|---|---|---|---|---|
| SparseInst | 8.6 | 0.1 | 6.6 | 8.3 | 8.7 | 10.8 | 26.7 | 36.9 | 13.3 |
| Mask2Former | 25.0 | 2.2 | 8.4 | 12.1 | 13.6 | 15.2 | 33.5 | 23.8 | 16.7 |
| MaskDINO | 41.0 | 13.1 | 22.9 | 35 | 36.7 | 40.1 | 46.5 | 45.6 | 35.1 |

We evaluated three methods on ReSOS(Table 6), where MaskDINO achieved the highest Average Precision (AP). Results show large objects segment better, while small vehicles and helicopters remain most challenging, underscoring the need for improved small object segmentation.

## 5 CONCLUSION

We proposed SOPSeg, a prompt-based framework with region-adaptive magnification, oriented prompts, and an edge-aware multi-scale refinement decoder for accurate small object segmentation in remote sensing imagery. Together with the new ReSOS dataset, SOPSeg provides a strong benchmark and resource to advance future research.

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
