# OpenReview forum: "SOPSeg: Prompt-based Small Object Instance Segmentation in Remote Sensing Imagery"
_ICLR.cc/2026/Conference — ICLR 2026 Conference Withdrawn Submission_

### Official Review · Reviewer_Umos · 2025-10-31

**Soundness:** 3
**Presentation:** 3
**Contribution:** 3
**Rating:** 6
**Confidence:** 4

**Summary:**

The paper tackles the challenge of small object instance segmentation in remote sensing imagery. This task remains largely unexplored due to the lack of pixel-level annotated datasets and the difficulty of segmenting fine-scale targets. To address this, the authors propose SOPSeg (Small Object Prompted Segmentation), a prompt-based framework that adapts the Segment Anything Model (SAM) for small-object segmentation. SOPSeg introduces three key innovations:
1)	Region-adaptive magnification to preserve fine spatial details lost during downsampling.
2)	An edge-aware decoder that combines boundary prediction and progressive refinement for precise mask generation.
3)	An oriented prompting mechanism tailored for rotated bounding boxes common in aerial imagery.
SOPSeg achieves superior performance over SAM and other prompt-based baselines on multiple datasets, including iSAID, NWPU-VHR10, and SAT-MTB. Furthermore, it is used to construct ReSOS, the first large-scale remote sensing small object instance segmentation dataset, containing over 709,000 pixel-level annotations derived from SODA-A. Overall, the work provides both a novel model and a valuable dataset, advancing research on fine-grained small object analysis in remote sensing.

**Strengths:**

The paper is original in focusing on small object instance segmentation in remote sensing imagery, a problem that has received limited attention. It proposes a reasonable and targeted framework (SOPSeg) that extends the Segment Anything Model with region-adaptive magnification, edge-aware decoding, and oriented prompting, effectively addressing the limitations of existing methods on small targets. The technical design is sound, and the experiments on several public datasets provide convincing evidence of improvement. The construction of the ReSOS dataset also adds practical value by supporting future research. Overall, the paper is clearly presented.

**Weaknesses:**

1)	Dataset construction lacks transparency.
While the ReSOS dataset is an important contribution, the paper provides limited information on annotation quality control and manual filtering criteria. Details such as the number of annotators, inter-annotator agreement, or filtering ratio would improve reproducibility and trust in dataset quality.
2)	Generalization evaluation could be expanded.
The model is validated on iSAID, NWPU-VHR10, and SAT-MTB, which share similar imaging styles and object categories. Testing on datasets with different sensors or resolutions (e.g., DOTA or FAIR1M) would better demonstrate robustness across varying data domains.
3)	 Computational efficiency discussion is limited.
Although the paper reports GFLOPs, there is no analysis of runtime or memory consumption under different region magnification settings. Since SOPSeg involves region extraction and multi-scale refinement, efficiency and scalability to large scenes remain unclear.
Overall, the work is technically solid and practically useful, but the dataset transparency, and analytical depth could be further improved to strengthen its scientific contribution.

**Questions:**

On the region-adaptive magnification strategy:
1)	Would it be possible to learn these parameters dynamically through a lightweight network or reinforcement signal rather than using a predefined rule?
On the oriented prompting mechanism:
1)	The method selects the object center and two midpoints of the short edges as prompts. How sensitive is the segmentation performance to the exact location or number of these points?
2)	In cluttered scenes where objects overlap, how does the prompting mechanism avoid interference between nearby instances?
On the edge-aware decoder and refinement process:
1)	Could the authors clarify how the edge branch interacts with the main mask branch during multi-scale refinement? Is there any feature sharing or fusion between them?
2)	What is the computational overhead introduced by the edge prediction and iterative refinement steps compared with the original SAM decoder?
3)	Have the authors visualized intermediate feature maps or edge activations to show how the refinement improves small-object boundaries?
On the ReSOS dataset construction:
1)	The dataset creation pipeline mentions automatic mask generation followed by manual filtering. Could the authors provide more details on the filtering process—such as the proportion of removed masks, annotator consistency, or validation criteria?
2)	How is annotation quality verified? Were any quality metrics (e.g., IoU between annotators, random spot checks) used?

---

### Official Review · Reviewer_i7qd · 2025-10-31

**Soundness:** 2
**Presentation:** 2
**Contribution:** 2
**Rating:** 4
**Confidence:** 5

**Summary:**

This paper addresses the challenging and underexplored task of small object instance segmentation in remote sensing imagery. The authors identify a key limitation in existing models like SAM: their coarse feature resolution leads to a significant loss of detail, hampering performance on small objects.
To overcome this, the paper introduces SOPSeg, a prompt-based framework with three main contributions:

1.**A novel model architecture:** It incorporates a region-adaptive magnification strategy to preserve fine-grained details and a customized decoder that uses edge prediction and progressive refinement for accurate boundaries.

2.**A tailored prompting mechanism:** It is designed to work with oriented bounding boxes, which are prevalent in remote sensing applications.

3.**A new public resource:** The authors construct and will release SODA-A-seg, a dataset for small object instance segmentation, addressing a resource gap in this domain.

**Strengths:**

The paper's main merit lies in identifying and tackling a significant and challenging problem: instance segmentation of small objects in remote sensing imagery. This is an important, practical area where foundational models like SAM are known to struggle, so the motivation is strong and well-founded.

The commitment to construct and release a new dataset, SODA-A-seg, is a laudable goal. The community lacks such a dedicated resource for small object instance segmentation, and if executed well, this could become a valuable contribution. Creating new benchmarks is a critical service to the field.

Conceptually, some of the proposed architectural ideas are sound. The region-adaptive magnification strategy is a logical approach to address the loss of fine-grained details for small objects, which is a known limitation of standard Vision Transformer backbones. Similarly, adapting the prompt mechanism to handle oriented bounding boxes (OBBs) demonstrates an awareness of the specific needs of the remote sensing domain.

**Weaknesses:**

While the paper addresses an important problem, its current execution suffers from several major weaknesses that undermine its contributions and call its conclusions into question.

1.**Insufficient Empirical Support for Core Claims:** The paper's central motivation—that SAM's failure on small objects is primarily due to its coarse 1/16 feature resolution—is presented as a fact but is never experimentally verified. A rigorous paper would include an analysis or ablation study to support this claim. For instance, the authors could have evaluated features from earlier, higher-resolution stages of SAM's encoder or compared its performance on small versus large objects within the same dataset to quantify the performance degradation.

2.**Inadequate Experimental Comparisons:** The baseline coverage is too narrow, as the main table compares almost exclusively with SAM series (SAM/HQ-SAM/ROS-SAM) and UGBS, lacking comparisons with non-SAM prompt-based frameworks. This narrow scope weakens the generality claim and raises fairness concerns around prompts; to substantiate generality, include strong non-SAM baselines (e.g., Mask2Former, MaskDINO, SparseInst).

3.**The Dataset Contribution is Unverifiable:** The paper claims the new SODA-A-seg dataset as a major contribution, yet it provides almost no details about it. The text explicitly defers these crucial details—such as dataset size, object categories, object size distribution, annotation protocol, and quality control measures—to an appendix that is not provided with the submission. This is a critical omission. Without this information, it is impossible for reviewers to assess the quality, scale, or significance of this claimed contribution. A contribution that cannot be reviewed is, for all practical purposes, non-existent in the context of this submission.

4.**Poor Presentation Quality and Lack of Clarity:** The overall presentation of the paper falls short of the standards expected at ICLR. There are numerous issues with the layout and formatting, such as the misalignment of Figure 4 and Table 5, which disrupt the flow and suggest a lack of careful preparation. Beyond aesthetics, many figures are poorly explained, lack detailed captions, and are difficult to interpret. This lack of clarity is not merely cosmetic.

**Questions:**

**Q1.** ReSOS masks are generated by SOPSeg then filtered manually. How did you quantify and mitigate method-induced bias/circularity—e.g., percent masks edited/rejected, cross-model re-annotation, and impact on non-SOPSeg methods’ rankings?

**Q2.** When compared with SAM/HQ-SAM/ROS-SAM/UGBS, are identical oriented prompts (OBB + keypoints) provided for all baselines? If not, please report the results after adding equivalent prompts to isolate the OPM gain and ensure fairness.

---

### Official Review · Reviewer_Mjck · 2025-10-31

**Soundness:** 3
**Presentation:** 3
**Contribution:** 2
**Rating:** 2
**Confidence:** 4

**Summary:**

This paper introduces a SAM-style instance segmentation framework for small objects in high-resolution remote sensing images. The proposed framework also extends a previous object detection samples to a new datasets with masks. The experiments on several datasets verified the effectiveness of the additional information extraction and support for the proposed framework.

**Strengths:**

The paper is easy-to-follow. The explanations on the proposed modules are clear. The experiments report the key metrics of instance segmentation work.

**Weaknesses:**

I have several questions and comments on this paper.

1. Why did the authors select 7 out of 15 categories? Are these categories are all small objects? How about the others? I cannot find a clear definition on the 'small' concept. The missing information probably makes it unclear for readers.
2. How is the generated data quality based on the SOPSeg framework? The authors claimed that they did the manual filtering to refine the samples. However, I cannot find any information about the details regarding the quality and filtering.
3. The authors pointed out previous works cannot handle diverse small objects. However, in the settings from this paper. the categories only cover 7 types. We cannot know the real performance when facing multiple and diverse small objects in remote sensing domain, e.g., panel, window, flag. or road sign.
4. From my understanding, the region-adaptive magnification strategy rely on the coarse bounding box annotation for the precise localization, then it's not fair to directly compare the results with other models as it has additional information fed into the network. Moreover, finding a small object from remote sensing images is not easy, either. which makes the motivation not so convincing.
5. The two prompt points are generated based on the bounding boxes, while the orientation of a object is not always consistent with its mask prediction. They are not 100% correct for a complex object.
6. What is the motivation of the evaluation 4.4? We can know the coarse performance of the popular models on the proposed dataset. However, we don't know the low performance is caused by the data complexity or the restriction of the models on the remote sensing images, or the correctness of the data.
7. The proposed strategies are not new in remote sensing domain, e.g., local cropping, point-orientation, and the progressive refinement.

**Questions:**

See weaknesses above.

**Details Of Ethics Concerns:**

The previous datasets modified in this paper.

---

### Official Review · Reviewer_Q44A · 2025-11-03

**Soundness:** 3
**Presentation:** 3
**Contribution:** 3
**Rating:** 6
**Confidence:** 3

**Summary:**

This paper proposes a new model named SOPSeg for small object instance segmentation of remote sensing images, and produces a new large-scale remote sensing small object instance segmentation dataset ReSOS. Based on the original architecture of SAM, the model introduces an edge-aware decoder, adds a parallel edge prediction branch, and gradually refines the prediction results of the model by combining multi-scale features to improve the instance segmentation effect of small objects. In order to cope with the loss of small object information caused by the downsampling operation in the image encoder, the Region-adaptive magnification(RAM) strategy increases the size of the small object in the input image by cropping and amplifying the size of the region where the small object is located. At the same time, the oriented prompting mechanism(OPM) is used to give the model directional point prompts to cope with the rotating bounding box in the aerial image. Experiments show that the method achieves sota performance.

**Strengths:**

1.The SOPSeg proposed by the author has achieved the best performance compared with other prompt-based methods on multiple datasets.
2.Sufficient ablation experiments verified the performance improvement brought by each module.
3.The author has constructed a new remote sensing image small target instance segmentation data set to provide support for subsequent research on related tasks.

**Weaknesses:**

1.This paper only compared the model performance with the prompt-based model in the experiment of model performance comparison.
2.The performance improvement of the SOPSeg mainly comes from the technical processing operations of the two modules of RAM and OPM, rather than the innovation of the model.

**Questions:**

1.The author seems to have only compared the model performance with the prompt-based model in the experiment. Can you consider adding some non-this type of model for performance comparison?
2.In the region adaptive amplification strategy, how to select the object for the cropping and amplification of the small target dense region?
3.In the region adaptive amplification strategy, if the complete target object in the original image appears incomplete at the edge of the cropping region, how to solve the adverse effects on the model?
4.When the target is no longer an elongated object such as a vehicle and a ship, but an object with an irregular shape, can the midpoint of the short-edge in the oriented prompting mechanism still provide accurate object direction guidance?

---

### Note · Authors · 2025-11-14

I have read and agree with the venue's withdrawal policy on behalf of myself and my co-authors.